# A Dual-Protection Framework for Copyright Protection and Image Editing Using Multi-Label Conformal Prediction

## Abstract

Recent advances in diffusion models have significantly enhanced image editing capabilities, raising serious concerns about copyright protection. Traditional watermarks often fail to withstand diffusion-based edits, making image protection challenging. To address this, we propose a method that embeds an imperceptible perturbation in images, serving as a watermark while simultaneously disrupting the output of latent diffusion models. Our method employs a Score Estimator trained on select latent embeddings to embed the watermark by minimizing the score function. We then apply conformal inference to compute p-values for watermark detection. To distort the output of latent diffusion models, we shift watermarked image embeddings away from the distribution mean, distorting unauthorized generations. Experiments demonstrate our framework's superior performance in watermark detection, imperceptibility, and distortion efficacy, offering a comprehensive approach to protect images against latent diffusion models.

## 1 Introduction

Traditional methods for protecting image copyrights often rely on embedding imperceptible messages as digital watermarks into images (Zhu et al., 2018). These watermarks allow creators to verify ownership by detecting their presence in suspected unauthorized copies. Although effective against direct misuse, such approaches face limitations with the emergence of generative models (Goodfellow et al., 2014), particularly diffusion models (Dhariwal & Nichol, 2021). By learning the underlying distribution of a given dataset, diffusion models can produce novel images that closely resemble the training data (Song et al., 2020), which raises serious copyright concerns. Existing watermarks face two unique challenges: first, feeding watermarked images into diffusion pipelines (e.g., for editing or style transfer) can degrade or distort embedded watermarks in the outputs, complicating detection (Mareen et al., 2024). Second, malicious users can exploit diffusion models to generate new images using watermarked images as input, undermining copyright protection efforts.

These vulnerabilities threaten creators' rights and creative integrity, for example, having their original works replicated or modified without permission. Traditional watermarking techniques that focus on embedding invisible information within images fail to address this challenge: they neither prevent the generation of new images or the modification of the original using diffusion models, nor ensure watermark robustness against diffusion-based generation processes (Rombach et al., 2022).

In this work, we propose a novel dual-protection framework designed to address both image watermarking and the prevention of misuse by latent diffusion models (LDM) (Rombach et al., 2022). Our approach introduces an adversarial perturbation that acts as a watermark by leveraging a Score Estimator trained on the latent embeddings generated by an LDM encoder (Esser et al., 2021) (Figure 1). The watermark is made invisible by constraining the perturbation strength to ensure minimal perceptual change. To embed the watermark in the latent space, we employ conformal inference (Angelopoulos & Bates, 2021; Tibshirani, 2023) together with adversarial optimization to minimize the conformity score, driving it to a value that statistically should occur with a probability of less than the significance level. To verify the presence of a watermark, we calculate p-values and compare them against a pre-determined threshold, providing a rigorous guaranty for controlling type I errors. Additionally, the adversarial perturbations are designed to shift the latent embedding of wa-

Figure 1: Existing methods that embed a predefined key as watermark cannot prevent malicious users from using these images to re-generate or train a latent diffusion model (LDM). Our method generates unique watermarks using adversarial optimization that minimizes a loss over the latent space and a conformity score. We shift the watermarked image's distribution to a low sampled region of the LDM's latent space which prevents the LDM to further train or sample from it, resulting in distorted outputs. Moreover, we can detect the presence of our watermark using a statistical test satisfying our dual protection goal.

termarked images away from the mean of the embedding distribution, placing them in low-density regions. As a result, when a LDM tries to use this image as input, it forces the model to generate visibly distorted outputs, ensuring detectable artifacts while maintaining watermarking objectives. The key contributions of our work are as follows:

- We propose a dual-protection framework that introduces an invisible perturbation to images, serving as a watermark while simultaneously distorting the output of LDMs.
- Our method leverages conformal inference to calculate p-values for watermark detection, providing a statistically robust approach to identifying watermarked images.
- Unlike previous watermarking techniques, our framework is designed to prevent malicious users from directly claiming ownership of a watermarked image using its embedding (Case 2 of Section 5.7). By utilizing conformal inference, the selection of watermark dimensions remains hidden, making any malicious claim of ownership no better than a random guess. This significantly strengthens the defense against unauthorized ownership assertions.

In summary, our approach provides a comprehensive solution for copyright protection against LDMs. By embedding invisible perturbations that act as both watermarks and deterrents to misuse, we offer a novel mechanism for protecting creative works while ensuring that LDMs cannot be easily exploited to generate unauthorized content.

## 2 RELATED WORK

**Adversarial Attacks** In computer vision, adversarial examples are subtly altered images that manipulate neural network models while remaining nearly imperceptible (Szegedy et al., 2013). In image classification, these examples can cause models to misclassify images. In diffusion models, adversarial attacks introduce undetectable changes that result in distorted outputs, revealing tampering. Salman et al. (2023) identify two types of adversarial attacks on LDMs: encoder attacks, which modify the encoder's output to resemble a predefined embedding, and diffusion attacks, which match the model's output to a target image. Glaze (Shan et al., 2023) protects artists' styles from unauthorized replication by incorporating invisible perturbations to distort styles learned by diffusion models .

**Image Watermarking** Image watermarking embeds invisible information in images to assert copyright (Cox et al., 2007). With the rise of diffusion models capable of producing high-quality images, traditional digital watermarking techniques, like HiDDeN (Zhu et al., 2018) and SSL (Fernandez et al., 2022) face new challenges. HiDDeN is an end-to-end CNN framework comprising an encoder, decoder, and adversarial network. SSL leverages self-supervised learning to embed watermarks in pretrained latent spaces. Tree-Ring (Wen et al., 2023) instead embeds imperceptible patterns into the initial noise vectors of diffusion models through Fourier-space structuring. Meanwhile Stable Signature (Fernandez et al., 2023) fine-tunes the decoder of LDMs to natively embed watermarks. Secret Key Signature (SKS) (Chen et al., 2024) uses adversarial attacks to embed watermarks into images, accompanied by hypothesis tests for detecting watermarks with statistical guarantees. In this work, we aim to achieve comparable watermarking performance while distorting the outputs

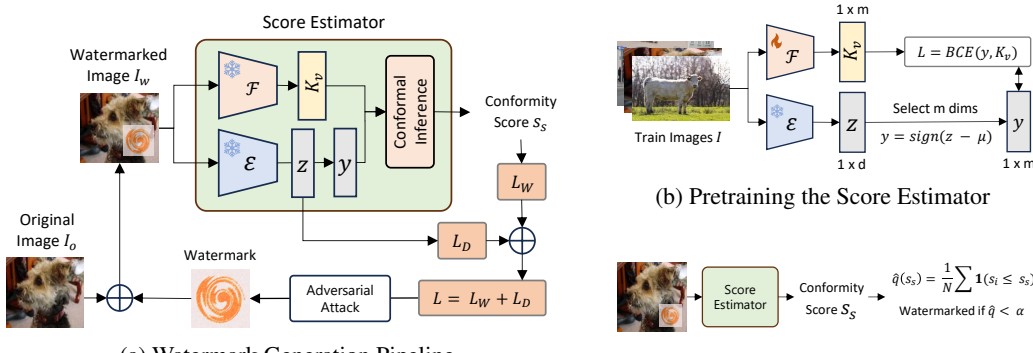

(a) Watermark Generation Pipeline

(b) Pretraining the Score Estimator

(c) Watermark Detection

Figure 2: Pipeline of our proposed method. (a) The input image $I_o$ is passed through a feature embedding model $\mathcal{F}$ to obtain the key vector $K_v$ and a latent diffusion model encoder $\mathcal{E}$ to get the latent vector $z$. The label $y$ is calculated using Equation 2. We use conformal inference to generate a score $s_s$ of the key vector. The watermark is a perturbation embedding generated by minimizing the loss over $s_s$ and $z$. (b) Prior to watermark generation, we pretrain $\mathcal{F}$ using a large set of images in order to generate key vectors which are drawn from the mean distribution of the dataset. (c) To detect whether an image is watermarked or not, we calculate a p-value based on the score. If the value falls below a certain threshold $\alpha$ we can claim that the image is watermarked.

of LDMs. Unlike previous watermark-only methods, such as SKS, our approach protects watermarked images from manipulation by LDMs. Furthermore, by incorporating conformal inference, our method resists direct ownership claims—an issue that SKS cannot address.

## 3 METHODS

This section describes our dual-protection strategy – (i) integrate an invisible watermark in images, and (ii) distort the output of LDMs that attempt to utilize these images without authorization.

As shown in Figure 2, we first train a Score Estimator to predict a conformity score $s_s \in \mathbb{R}$ for an input image $I_o$. Next, using the trained estimator, we run a joint optimization loop to add a small perturbation to $I_o$ to produce a watermarked image $I_w$. The joint optimization needs to balance between the watermark strength to ensure imperceptibility as well as to change the distribution of the original input so that it becomes distorted when used by an LDM. This is achieved by minimizing:

$$L(I_o) = \lambda_{\mathrm{D}} L_{\mathrm{D}}(I_o) + \lambda_{\mathrm{W}} L_{\mathrm{W}}(I_o), \tag{1}$$

where $L_{\mathrm{D}}$ represents the loss that distorts the output of the LDM, and $L_{\mathrm{W}}$ represents the loss for embedding the watermark. The parameters $\lambda_{\mathrm{D}}$ and $\lambda_{\mathrm{W}}$ are adjustable weights. We optimize $L(I_o)$ using a modified momentum-based iterative algorithm, MI-FGSM (Dong et al., 2018), adapted with a parameter $\beta_{\mathrm{tg}}$ to bound the perturbation's magnitude. A larger $\beta_{\mathrm{tg}}$ relaxes this constraint, permitting stronger perturbations. The detailed steps of MI-FGSM are provided in the appendix.

In the watermark detection stage, to determine whether a suspected image belongs to the image owner, the image is input into the Score Estimator. A p-value is calculated via conformal inference and a reference set of calibration images. If the value falls below the confidence level, we can assert that the image is watermarked.

### 3.1 WATERMARK EMBEDDING

In this section, we describe the process of embedding a watermark into an image. Specifically, we elaborate on the definition of $L_{\mathrm{W}}$ in equation 1.

#### 3.1.1 SCORE ESTIMATOR MODEL

We first introduce the Score Estimator, which contains a CNN-based feature embedding model $\mathcal{F}$ that generates $m$-dimensional binary key vectors aligned with the indices of the latent space of a

frozen LDM encoder $\mathcal{E}$. To ensure watermark security, we generate unique labels instead of relying on known labels. Training data is created by sampling an image $I$ from a large dataset and passing it through $\mathcal{E}$ to obtain a latent embedding $z = \mathcal{E}(I) \in \mathbb{R}^d$. We then compare $z$ with the dataset latent embedding mean $\mu$ and compute the sign of this difference: $\text{sign}(\mathcal{E}(I) - \mu)$. From this sign vector, $m$ dimensions are randomly selected as the binary label $y \in \{-1, 1\}^m$, i.e,

$$y = \text{sign}(\mathcal{E}(I) - \mu)_{i_1, i_2, \dots, i_m}, \tag{2}$$

where $i_1, i_2, \dots, i_m$ are randomly selected dimensions chosen by the image owner. The same set of $m$ dimensions are used consistently across all images for both training and watermarking. Since the total number of dimensions $z$ is large, it is computationally infeasible for malicious users to identify the chosen subset selected by the user, thereby protecting the watermark's privacy.

Next, we train $\mathcal{F}$ to output the $m$-dimensional vector. A sigmoid activation function is applied to the model output to produce probabilities for each dimension. The predictor model is trained using binary cross-entropy loss for multi-label classification on a large image dataset with our generated keys, shown in Figure 2b. We refer to this trained model as the Score Estimator. During watermark generation and detection, conformity scores are computed using multi-label conformal prediction (Cauchois et al., 2021) utilizing both the generated keys and latent space labels.

### 3.1.2 MULTILABEL CLASSIFICATION CONFORMAL INFERENCE

We adopt the multilabel classification conformal inference method proposed by Cauchois et al. (2021), which forms the foundation of our watermark approach. Conformal inference is a statistical framework that provides valid confidence levels for predictions (Tibshirani, 2023). In the context of multilabel classification, it computes a conformity score for each label and constructs a prediction set containing the true labels with a predefined confidence level. Given an image $I \in \mathcal{I}$ and its corresponding $k$-th label $y_k \in \{-1, 1\}$, our goal is to compute an overall conformity score for the image by considering dependencies among multiple labels. Cauchois et al. (2021) proposes building a tree structure to capture these dependencies and then compute the conformity score based on it.

We define two factors to model label dependencies: interaction factors and marginal factors. The interaction factors $\psi : \{-1, 1\}^2 \to \mathbb{R}^4$ capture pairwise label interactions, defined as:

$$\psi(-1, -1) = e_1, \ \psi(1, -1) = e_2, \ \psi(-1, 1) = e_3, \ \psi(1, 1) = e_4 \tag{3}$$

where $e_1, e_2, e_3, e_4$ are the standard basis vectors of $\mathbb{R}^4$. Meanwhile, marginal factors $\phi_k : \{-1, 1\} \times \mathcal{I} \to \mathbb{R}^2$ describe how the individual labels relate to $I$. The marginal factor for the $k$-th label is:

$$\phi_k(y_k, I) := \frac{1}{2} \begin{pmatrix} (y_k - 1) \cdot s_k(I) \\ (y_k + 1) \cdot s_k(I) \end{pmatrix} \tag{4}$$

where $s_k(\cdot)$ is the score function for the $k$-th label. To model the dependencies among labels, a tree-structured graphical model (Chow & Liu, 1968) is employed. For a tree $\mathcal{T} = ([K], E)$, the joint probability of the labels is modeled as:

$$p_{\mathcal{T}, \alpha, \beta}(y \mid I) \propto \exp\left(\sum_{e=(k,l) \in E} \beta_e^T \psi(y_k, y_l) + \sum_{k=1}^K \alpha_k^T \phi_k(y_k, I)\right) \tag{5}$$

where $\alpha$ and $\beta$ are parameters that describe the interaction and marginal contributions, respectively. Specifically, $\alpha_k \in \mathbb{R}^2$ for each label $k$, and $\beta_e \in \mathbb{R}^4$ for each edge $e \in E$.

The tree structure $\mathcal{T}$ that best represents the dependencies between labels is learned by maximizing the log-likelihood of training data. Given a training dataset $\mathcal{D}_{\text{train}} = \{(I^{(i)}, y^{(i)})\}_{i=1}^{N_{\text{train}}}$, we optimize

$$\hat{\mathcal{T}}, \hat{\alpha}, \hat{\beta} = \arg\max_{\mathcal{T}, \alpha, \beta} \sum_{i=1}^{N_{\text{train}}} \log p_{\mathcal{T}, \alpha, \beta}(y^{(i)} \mid I^{(i)}) \tag{6}$$

To estimate the dependencies between labels, we compute the empirical mutual information between each pair of labels using single-edge trees. The optimal tree is obtained by solving for the maximum spanning tree based on mutual information values (Chow & Liu, 1968). With the tree structure and parameters learned, we define a scoring function $s(I, y)$ for an image $I$ and its label set $y$, given by:

$$s_{\hat{\mathcal{T}}, \hat{\alpha}, \hat{\beta}}(I, y) := \sum_{e=(k,l) \in \hat{E}} \hat{\beta}_e^T \psi(y_k, y_l) + \sum_{k=1}^K \hat{\alpha}_k^T \phi_k(y_k, I) \tag{7}$$

### 3.1.3 WATERMARK EMBEDDING LOSS

Using the trained key generator $\mathcal{F}$ and the latent labels $y$, we create a watermark via conformal inference. The scoring function between the $k$-th output $\mathcal{F}(I)_k$ and the $k$-th label $y_k$ is defined as:

$$s_k(I, y) = -|\mathcal{F}(I)_k - y_k|. \tag{8}$$

Using multi-label conformal prediction, we estimate the edge empirical mutual information for every pair of nodes in the $m$ dimensions selected, and construct a tree-structured score function $\hat{s}(I, y)$ based on the Chow-Liu-type approximate maximum likelihood tree. Next, we compute conformity scores for a calibration set by passing them through $\mathcal{F}$ to generate keys and evaluating $\hat{s}(I, y)$. Using these scores, we determine an empirical critical value $s_{\mathrm{cv}}$. An image is classified as watermarked if its conformity score satisfies $\hat{s}(I, y) < s_{\mathrm{cv}}$. To embed a watermark, we optimize the original image $I_o$ to minimize its score such that $\hat{s}(I_o, y) \leq s_{\mathrm{cv}}$, yielding the watermarked image $I_w$.

In practice, when minimizing the scoring function $\hat{s}(I, y)$ with respect to $I$, we must account for the gradient passing through $y$: $\frac{\partial \hat{s}}{\partial y} \frac{\partial y}{\partial I}$. However, since $y$ is discrete, $\frac{\partial y}{\partial I}$ is zero. To address this, we "soften" $y$ using the sigmoid function $\sigma$, such that $\tilde{y} = \sigma(y)$. Therefore, the loss function for watermark embedding is:

$$L_{\mathrm{W}}(I_o) = \hat{s}(I_o, \tilde{y}). \tag{9}$$

## 3.2 DISTORTING LATENT DIFFUSION MODELS

The last part of our objective function is the distortion loss $L_{\mathrm{D}}$, designed to distort the latent diffusion model's output. For an input $I_o$, we shift its latent embedding $\mathcal{E}(I_o)$ into low-probability regions of the latent space. These regions correspond to under-sampled patterns during the diffusion model's training. This forces the diffusion process to operate outside its learned manifold, inducing higher denoising errors and distortions in generated outputs.

As demonstrated later in the appendix, the distribution of image embeddings $z$ within the latent space $\mathcal{Z}$ follows a Gaussian distribution $f(z)$, characterized by a mean vector $\mu \in \mathbb{R}^n$ and a covariance matrix $\Sigma \in \mathbb{R}^{n \times n}$. Given $I_o$, our objective is to create a watermarked image $I_w$ by minimizing the log-likelihood of the image embedding $z_w$. We minimize the following loss with respect to $I_o$,

$$L_{\mathrm{D}}(I_o) = -\frac{1}{2}(\mathcal{E}(I_o) - \mu)^\top \Sigma^{-1}(\mathcal{E}(I_o) - \mu). \tag{10}$$

# 4 WATERMARK DETECTION

We test whether a suspected image $I_s$ is watermarked using two hypotheses: $H_0$ (null) states that $I_s$ is not watermarked, and $H_1$ (alternative) states that $I_s$ is watermarked. To test for the presence of a watermark, we assume access to a calibration dataset $\mathcal{D}_{\mathrm{cal}} = \{(I_i, y_i)\}_{i=1}^N$, where $I_i$ is the $i$-th image, $y_i$ the corresponding label calculated using equation 2, and $N$ is the calibration dataset size. This calibration dataset is distinct from the one used in Section 3.1.3 to maintain statistical validity of the hypothesis test. For each calibration image $I_i$, we compute its conformity score $s_i = \hat{s}(I_i, y_i)$ and sort these scores in ascending order. Using the same feature dimensions $\{i_1, \ldots, i_m\}$ selected during watermark embedding, the suspected image's conformity score is computed as $s_s = \hat{s}(I_s, y_s)$.

The empirical critical value is then calculated as $s_{\mathrm{cv}} = s_{(\lceil(N+1)\alpha\rceil)}$, where $\lceil \cdot \rceil$ denotes the ceiling function and $\alpha$ is the desired significance level. We reject $H_0$ if $s_s < s_{\mathrm{cv}}$, indicating the presence of a watermark. We compute the p-value of the suspected image under hypothesis testing as:

$$\hat{q}(s_s) = \frac{1}{N} \sum_{i=1}^N \mathbf{1}(s_i \leq s_s). \tag{11}$$

# 5 EXPERIMENT

We evaluate our method in six aspects: detection performance, imperceptibility, distortion analysis, generalization, robustness, and security.

## 5.1 EXPERIMENT SETUP

Our experiments use the MSCOCO 2017 dataset (Lin et al., 2014) containing 118k training, 41k test, and 5k validation images. We adopted VGG16 (Simonyan, 2014) as the backbone for our Score Estimator to generate $m$-dimensional key vectors. During experiments, we found VGG16's moderate accuracy (compared to ResNet (He et al., 2016)) provides better uncertainty calibration for conformal inference-based watermark embedding. We use the encoder of Stable Diffusion as $\mathcal{E}$ to generate latent embeddings and train the VGG model using binary cross-entropy loss (Equation 1). Detailed training configurations are provided in the appendix.

We optimize Equation 1 using MI-FGSM (Dong et al., 2018), evaluating performance in three aspects: (1) detection rate, measured as the percentage of watermarked images with p-values below the significance level; (2) image quality, quantified by PSNR, SSIM, MAE, and RMSE; and (3) distortion, evaluated via FID between LDM outputs and reference WikiArt images (Tan et al., 2019).

We compare against three adapted baselines: SSL (Fernandez et al., 2022) in zero-bit mode ($\alpha = 0.05$), HiDDeN (Zhu et al., 2018) with a bit error rate threshold of 0.05, and SKS (Chen et al., 2024). All methods are modified by adding the distortion loss $L_{\mathrm{D}}$ to the original watermark loss, and use identical MSCOCO training/validation splits for fair comparison. For methods such as Stable Signature (Fernandez et al., 2023), which employ in-diffusion watermarking, watermarks are embedded into outputs of diffusion models. In contrast, our method embeds watermarks into images themselves, making a direct comparison of watermark properties unsuitable. Additionally, metrics like PSNR are computed based on the diffusion model's output. Since one of our primary objectives is to distort the output of LDMs, a direct comparison with these methods is not feasible.

Our experiments show that at equivalent image qualities, our method achieves comparable detection rates (99.94%) while inducing significantly stronger LDM distortion (FID 108.65 vs 72.68–92.35).

## 5.2 DETECTION PERFORMANCE ANALYSIS

We evaluated our watermark detection method by analyzing p-values for all watermarked images, computing their mean, standard deviation, and the percentage below significance levels of 0.05 and 0.01. To assess false positives, we perform the same analysis on clean images. As shown in Table 1, our method achieves near-perfect detection on watermarked images ($> 99\%$) with low mean p-values, while clean images exhibit false positive rates consistent with the chosen significance levels.

Table 1: Watermark detection performance: the mean and standard deviation (Std) of p-values, and the percentage below significance levels 0.05 and 0.01 for watermarked and clean images.

Table 2: FID comparison of the distortion introduced in LDM outputs across different methods. Higher FID indicates more distortion, offering better protection.

|  | Mean | Std | <0.05 | <0.01 |
|---|---|---|---|---|
| Clean | 0.4950 | 0.2871 | 4.6% | 0.92% |
| Watermarked | 0.0013 | 0.0039 | 99.94% | 99.52% |

| Method | Orig. | SSL | HiDDeN | SKS | Ours |
|---|---|---|---|---|---|
| FID↑ | 66.97 | 72.68 | 92.35 | 80.11 | **108.65** |

## 5.3 DISTORTION ANALYSIS

To measure the level of distortion introduced by different methods, we used 5,000 images from the MSCOCO validation set and generate watermarked images using different models. Next we input these watermarked images to the LDM with the prompt *Generate an image in the impressionism style of the original image*. We then compared the generated images with authentic Impressionist-style images from the WikiArt dataset (Tan et al., 2019), and compute FID as a measure of distortion.

This addresses a critical challenge in image protection: preventing LDMs from replicating the styles of artists. A higher FID indicates greater distortion in generated images, suggesting stronger protection against unauthorized generation. As shown in Table 2, our method introduces more distortion compared to other methods, providing better protection. Figure 3 compares the original image with various generated images: unwatermarked inputs yield high-quality Impressionism outputs closely matching prompts, whereas our watermarked images produce outputs with visible artifacts and blur. Additional results for Impressionism and other art styles are available in appendix.

Output from a Latent Diffusion Model

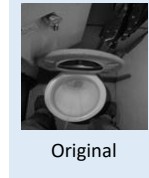 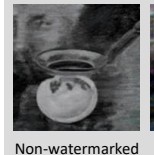 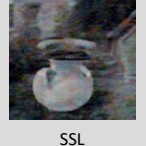 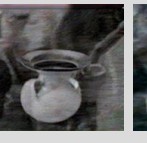 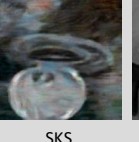 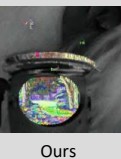

| Original | Non-watermarked Generation | SSL | HiDDeN | SKS | Ours |

Figure 3: Comparison of the original image with generated outputs using different watermarked images as input to a LDM with the prompt - *Generate an image in the impressionism style of the original image*. Our method produces the highest distortion in the generated result.

## 5.4 IMPERCEPTIBILITY ANALYSIS

We assess watermark imperceptibility by comparing the quality of watermarked images to the originals. HiDDeN achieved a PSNR of 33.56, and for fair comparison, SSL and SKS were adjusted to produce a similar PSNR of 33. Despite superior distortion effects, our method maintains better image quality across all metrics, as shown in Table 3. Additionally, by increasing $\beta_{\text{tg}}$ to allow greater perturbation in watermarked images, we matched the baseline PSNR (32) while achieving a FID of 150.65 and a 99.74% detection rate at $\alpha = 0.05$, demonstrating that stronger perturbations enhance LDM distortion. Visual examples of watermarked images and perturbations are shown in Figure 4.

Table 3: Comparison of watermark imperceptibility across different methods.

| Method | PSNR↑ | SSIM↑ | MAE↓ | RMSE↓ |
|--------|-------|-------|------|-------|
| HiDDeN | 32.72 | 0.9214 | 0.0242 | 0.0428 |
| SSL | 33.58 | 0.9408 | 0.0168 | 0.0222 |
| SKS | 32.01 | 0.9405 | 0.0165 | 0.0251 |
| Ours | **37.30** | **0.9412** | **0.0109** | **0.0195** |
| Ours (32) | 32.92 | 0.9147 | 0.0200 | 0.0278 |

Original      Watermarked      Perturbation x10

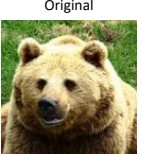 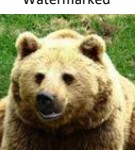 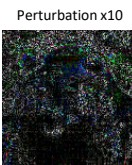

Figure 4: Visual comparison of original images, watermarked images, and 10× amplified structural differences.

## 5.5 GENERALIZATION ANALYSIS

We evaluate the generalization of our method across different LDMs by addressing two questions:

**Case 1: Can watermarked images trained on one diffusion model withstand attacks from another?** We trained the Score Estimator on Stable Diffusion 2 (SD2) (Ramesh et al., 2022) and input the watermarked images into Stable Diffusion XL (SDXL) (Podell et al., 2023) and DiffEdit (Couairon et al., 2022). We compared the FID scores of their outputs against those of the original images, as shown in Table 5. The results indicate that watermarked images trained for SD2 do not effectively distort other models. This is because the watermark relies on the latent encoder, and different LDMs use different encoders, limiting cross-model generalization.

**Case 2: Can the entire watermark pipeline transfer to another diffusion model?** We retrained the Score Estimator for InstructPix2Pix (Brooks et al., 2023) to generate watermarked images and evaluated the watermark detection rate, image quality, and FID (Table 4). Our method achieved a 98.84% detection rate at $\alpha = 0.05$ while maintaining superior image quality compared to baselines (Table 3). Furthermore, it significantly distorted InstructPix2Pix outputs, as reflected in higher FID.

Table 4: Image quality and FID comparison for watermark generated using InstructPix2Pix encoder.

| Method | PSNR | SSIM | MAE | RMSE | Method | Original | Watermarked |
|--------|------|------|-----|------|--------|----------|-------------|
| InstructPix2Pix | 36.38 | 0.9602 | 0.0119 | 0.0203 | FID | 96.73 | 132.79 |

In conclusion, watermarked images trained on one diffusion model do not effectively resist attacks from another model. However, our method generalizes well to different diffusion models when trained from scratch, demonstrating adaptability and robustness across architectures.

Table 5: FID comparison between original and SD2-watermarked images after processing by different models.

| Model | Original | Watermarked |
|---|---|---|
| SDXL | 117.54 | 122.69 |
| DiffEdit | 99.41 | 94.43 |

Table 6: Robustness test against multiple overlapping watermark embeddings. Detection rate of Alice's original watermark after embedding Bob's additional watermarks.

| | 1 | 2 | 3 | 4 | 5 |
|---|---|---|---|---|---|
| SSL | 92.62% | 60.50% | 27.00% | 19.50% | 12.00% |
| SKS | 99.50% | 98.60% | 96.10% | 89.70% | 89.70% |
| Ours | 99.94% | 96.52% | 92.80% | 84.62% | 75.46% |

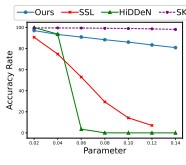 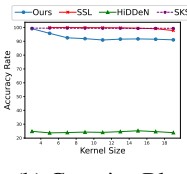 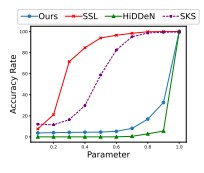 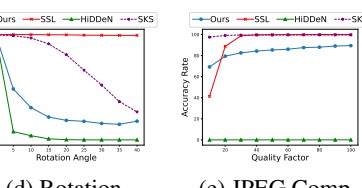

(a) Gaussian Noise   (b) Gaussian Blur   (c) Cropping   (d) Rotation   (e) JPEG Comp.

Figure 5: Comparison of the robustness of different watermarking methods under various image perturbations. Our method shows strong robustness under Gaussian Noise and JPEG Compression while being outperformed by SSL under some spatial transformations. Notably, our method achieves the highest resilience against diffusion model editing.

## 5.6 ROBUSTNESS ANALYSIS

To assess robustness, we tested our watermarking method under common image perturbations: Gaussian noise, Gaussian blur, cropping, rotation, and JPEG compression. We then measured the watermark detection rate (Figure 5). Our method demonstrates strong robustness under Gaussian noise, outperforming both HiDDeN and SSL. Besides, it consistently exceeds HiDDeN across all perturbations. However, SSL and SKS surpass our method in Gaussian blur, cropping, rotation, and compression. We attribute this difference to their higher watermark embedding dimensionality (2048 for SSL, 32 for SKS, vs. 6 for ours), which improves tolerance to spatial transformations.

For fairness, we retrain all baselines with the distortion loss $L_{\mathrm{D}}$, which enforces resistance to LDM-based editing but reduces robustness to perturbations. This suggests that simply combining standard watermark loss with distortion loss is not a good solution to achieve both aims. This highlights our method's advantage: we can achieve both aims while maintaining robustness against perturbations.

## 5.7 SECURITY ANALYSIS

We evaluated the security of our watermarking framework by simulating three attack scenarios involving two users: Alice, the image owner, and Bob, a malicious user attempting to claim ownership. We compared our method against other 0-bit watermarking approaches, SSL and SKS. For Cases 1 and 3, we took the results from Chen et al. (2024) without adding the distortion loss $L_{\mathrm{D}}$.

**Case 1: Fake Watermark Generation** Bob attempts to generate fake watermarks on clean images, hoping to bypass Alice's watermark detection. To simulate this, we randomly selected $m$ dimensions from the latent embedding $z$, trained a corresponding Score Estimator, and computed conformity score parameters in equation 7. We then watermarked the image using the loss in equation 1. Afterward, we evaluated the watermark detection rate using Alice's model to determine the effectiveness of Bob's attack. As shown in Table 7, our method yields a higher detection rate than SKS, though still lower than SSL. Notably, our method achieves a baseline detection rate comparable to SKS at a p-value threshold of 0.01. When applying the same threshold to evaluate Bob's attack, our detection rate improves to 1.88%—surpassing SKS's performance.

Table 7: Detection rate for fake watermark generation attacks across different methods.

| Method | SSL | SKS | Ours(0.05) | Ours(0.01) |
|---|---|---|---|---|
| Detection rate | 12.84% | 1.94% | 7.64% | 1.88% |

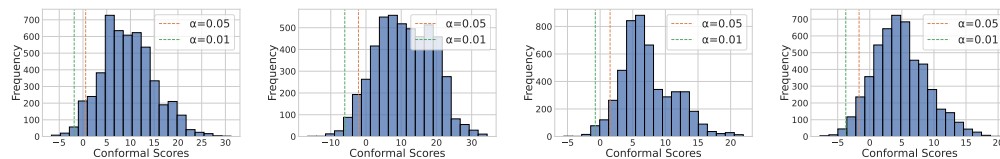

Figure 6: Conformity scores from four random dimension sets follow near-Gaussian distributions, making it difficult for Bob to directly train a Score Estimator to claim ownership of Alice's image.

**Case 2: Direct Ownership Claim** Bob tries to claim ownership of Alice's watermarked image without modifying it. To simulate this, we trained four Score Estimators with different randomly selected dimensions. Using the MSCOCO 2017 test set as a calibration set, we examined the conformity score distribution to determine whether Bob could achieve a p-value below 0.05. Since the conformity score distribution concentrates around the mean, Bob is unlikely to generate a valid watermark with a p-value below 0.05 without adversarial attacks. Figure 6 presents conformity score distributions for the four Score Estimators. In this scenario, SKS is vulnerable: Bob can take Alice's watermarked image and uses her SKN to generate a signature to claim ownership.

**Case 3: Watermark Removal and Replacement** Bob attempts to remove Alice's watermark by embedding his own. To test this, we embedded a watermark into an image (acting as Alice's watermark), then added another four watermarks as Bob's attack. We then checked whether Alice's watermark could still be detected after Bob's attack. Table 6 shows the detection rate of Alice's original watermark as more Bob's watermarks are added. SKS maintained a high detection rate (above 89%). Our method also showed resilience, with a detection rate of 84.62% after three additional watermarks, outperforming SSL. SSL's detection rate dropped significantly to 19.5% after three watermarks. Our slightly lower performance relative to SKS likely stems from the distortion loss $L_{\mathrm{D}}$, which affects all pixels, whereas the watermark loss $L_{\mathrm{W}}$ only affects a portion of the pixels (approximately 20%). This trade-off between detection and distortion explains the gap.

## 6    ABLATION STUDY

**Effect of Dimensionality in Feature Selection:** We analyze how the number of selected feature dimensions ($m$) affects performance. With the baseline configuration $m = 6$, FID = 108.65. Reducing to $m = 4$ lowers FID to 69.20, indicating weaker protection against unauthorized generation. The mean p-value for watermark detection is 0.0085, with 96.90% of p-values below 0.05 and 93.66% below 0.01, showing reduced detection robustness.

**Effect of Removing $L_{\mathrm{D}}$:** We tested the impact of removing the distortion loss by setting $\lambda_{\mathrm{SD}} = 0$. Without $L_{\mathrm{D}}$, our method achieved an FID 68.45 compared to 66.97 for the original images, showing only slightly more distortion. Surprisingly, the mean p-value for watermark detection was 0.0030, with 99.24% of p-values below 0.05 and 97.66% below 0.01. The detection rate performance was worse than when we included $L_{\mathrm{D}}$. We attribute this to the non-convex optimization in equation 1, where $L_{\mathrm{D}}$ acts as a small perturbation that helps escape local optima.

## 7    CONCLUSION

In this work, we propose a dual-protection framework to safeguard image copyrights against latent diffusion models. Our method embeds a perturbation into the image that serves both as a watermark and a mechanism for disrupting the outputs of latent diffusion models. Additionally, we leverage conformal inference to develop a statistically robust approach for detecting watermarked images. Experimental results demonstrate that our method ensures strong watermark detection, enhanced imperceptibility, and resilience against various image perturbations. Furthermore, we show that the entire pipeline generalizes across attacks involving different diffusion models. Notably, our approach resists direct ownership claims and multiple watermark embeddings, showing its potential as a reliable solution for protecting image copyrights in the era of generative AI.

## ETHICS STATEMENT

We propose a watermarking framework for latent diffusion models that embeds invisible watermarks for ownership verification and introduces adversarial perturbations to mitigate misuse. Our study uses only publicly available datasets (e.g., MS-COCO) under their licenses, with no human subjects or private data involved. The methods are designed to promote accountability and responsible use of generative models; while adversarial perturbations affect image quality, they serve solely as a protective measure against unauthorized exploitation.

## REPRODUCIBILITY STATEMENT

We have taken extensive measures to ensure the reproducibility of our work. All model architectures, training details, and hyperparameters are described in the main text and Appendix. Complete proofs of theoretical claims are provided in the supplementary material. We also detail dataset preprocessing steps and evaluation protocols to allow exact replication of experiments.

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

APPENDIX

## A  SMOOTHING FUNCTION FOR CONFORMAL INFERENCE

In the multi-label conformal inference method introduced by Cauchois et al. (2021), the function $\psi$ is defined in a discrete manner. Here, we propose a smoothing technique for this function. Specifically, we smooth the function $\psi(y_k, y_l)$ used in the multi-label conformal prediction set by defining:

$$\psi_c(y_k, y_l) = (1 - \pi_k)(1 - \pi_l) \cdot e_1 + (1 - \pi_k)\pi_l \cdot e_3 + \pi_k(1 - \pi_l) \cdot e_2 + \pi_k\pi_l \cdot e_4 \qquad (12)$$

where $\pi_k$ and $\pi_l$ are smoothing parameters, and $e_1$, $e_2$, $e_3$, and $e_4$ are the standard basis vectors.

Note that $\psi_c$ is equivalent to $\psi$ when $y_k, y_l \in \{-1, +1\}$. The corresponding loss function for the smoothed conformal prediction set is defined as:

$$L_c = \sum_{(k,l) \in \hat{E}} \beta_e^T \psi_c(y_k, y_l) + \sum_{k=1}^{K} \alpha_k^T \phi_k(y_k, x) \qquad (13)$$

where $\hat{E}$ denotes the set of edges in the maximum spanning tree, $\beta_e$ represents edge-specific parameters, and $\phi_k(y_k, x)$ is the marginal factor as defined in the aforementioned paper.

## B  MI-FGSM

To minimize the loss defined in equation 1, we employed an approach inspired by the MI-FGSM (Dong et al., 2018). For completeness, we outline the key aspects of this technique here. MI-FGSM is a widely used method for generating adversarial examples by iteratively adjusting the perturbation $\eta$ added to the input data (Dong et al., 2018). This adjustment aims to minimize the adversarial loss while ensuring that the perturbations remain minimal and imperceptible to the human eye. The update rule for the perturbation at each iteration $t$ is expressed as:

$$g_{t+1} = \mu g_t + \frac{\nabla_I L(f(I + \eta_t))}{\|\nabla_I L(f(I + \eta_t))\|_1} \qquad (14)$$

$$\eta_{t+1} = \eta_t - \alpha \cdot \text{sign}(g_{t+1}) \qquad (15)$$

In this equation, $L(f(I))$ represents the adversarial loss function, where $f(\cdot)$ is a deep neural network model, and $I$ is the input image. $g_t$ is the accumulated velocity vector in the gradient direction, initialized as $g_0 = 0$, and $\mu$ is the decay factor. The gradient $\nabla_I L$ is computed with respect to the input image $I$. To ensure the imperceptibility of the perturbations, a constraint on the magnitude of $\eta$ is typically enforced, such that $\|\eta_{t+1}\|_\infty < \epsilon$, where $\epsilon$ is a small threshold.

We made several modifications to MI-FGSM inspired by Chen et al. (2024) to better suit our specific problem:

1. **Direct Gradient Application:** Instead of using the sign of the gradient, we directly apply the gradient values to update the perturbation $\eta$, enhancing control over the perturbation process.

2. **Scaling Factor Introduction:** We introduced a scaling factor $\beta$ to expand the constraint $\|\eta_{t+1}\|_\infty < \epsilon$. The perturbation update is modified as follows:

$$\eta'_{t+1} = \beta \cdot \eta_{t+1} \qquad (16)$$

where $\beta$ is determined by the following formula:

$$\beta = \text{clip}\left(\sqrt{\frac{\beta_{\text{tg}}}{\text{mean}(\eta_{t+1}^2)}}, 0, 1\right) \qquad (17)$$

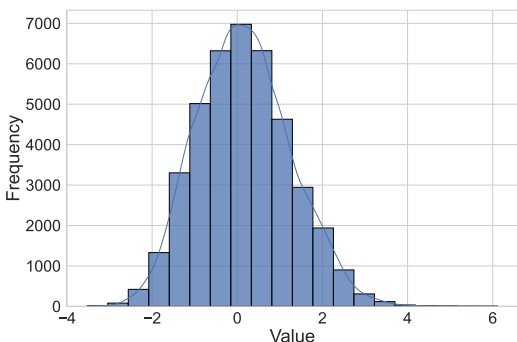

Figure 7: Distribution of one dimension of the embeddings from the latent diffusion model encoder.

Here, $\beta_{\text{tg}}$ is a target value similar to $\alpha$, serving to set an upper bound on the perturbation magnitude.

3. **Data Augmentation:** Before being input into the model $f(\cdot)$, the image $I$ undergoes data augmentation techniques, such as rotation and cropping, to improve the robustness of watermark detection. The updated rule becomes:

$$\eta_{t+1} = \eta_t - \alpha \cdot \nabla_I L(f(\text{da}(I)), y_{\text{target}}) \tag{18}$$

where $\text{da}(\cdot)$ denotes the data augmentation module.

4. **Adaptive Parameters:** Instead of using fixed parameters, we employ adaptive values for $\beta_{\text{tg}}$ and $\lambda_{\text{W}}$, allowing for a more flexible approach that enhances the success rate of watermark embedding.

These modifications to MI-FGSM make it more effective and suitable for our watermarking task, ensuring imperceptibility while achieving high watermark detection success.

## C    ASSESSING THE NORMALITY OF LATENT SPACE EMBEDDINGS

A critical assumption underlying our adversarial attack strategy is that the embeddings in the latent space follow a Gaussian distribution. To validate this assumption, we processed 118,000 images from the MSCOCO training set through the latent diffusion model encoder and analyzed the resulting embeddings using the Henze-Zirkler test for multi-dimensional normality. The test yielded a statistic of 0.0503, with a p-value of 1.0, allowing us to accept the null hypothesis that the latent space embeddings follow a Gaussian distribution. Figure 7 illustrates the distribution of one dimension of these embeddings.

## D    EXPERIMENT DETAILS

In our setting, we select the number of features for the watermark as $m = 6$. The step size is set to $\alpha = 0.1$, and the decay factor $\mu = 0.9$. For training, we use $\lambda_{\text{SD}} = 1$ and $\lambda_{\text{W}} = 1.6$. The scaling factor is initialized with $\beta_{\text{tg}} = 8 \times 10^{-4}$. For the adaptive parameters in MI-FGSM, after 200 iterations, we multiply $\lambda_{\text{W}}$ by 30 and $\beta_{\text{tg}}$ by 10. Additionally, after every 50 subsequent iterations, both parameters are multiplied by 3.

For training the VGG model, we use a learning rate of $1 \times 10^{-4}$, weight decay of $1 \times 10^{-4}$, batch size of 64, and apply a learning rate decay of 0.8 every 10 steps.

When training the conformality score Chow-Liu tree parameter, we use 5,000 images from the COCO test dataset. We then use an additional 5,000 images from the COCO validation dataset as the calibration set for computing the p-values.

---

**Algorithm 1** Score Estimator Pretraining

---

**Require:** Training images $I$, encoder $\mathcal{E}$, mean vector $\mu$, selected indices $\{i_1, \ldots, i_m\}$
**Ensure:** Trained CNN network $\mathcal{F}$
  1: **for** each image $I$ in dataset **do**
  2:     Compute binary label: $y \leftarrow \{\text{sign}(\mathcal{E}(I) - \mu)\}_{\{i_1, \ldots, i_m\}}$
  3: **end for**
  4: Initialize CNN network $\mathcal{F}$ with random weights
  5: **while** not converged **do**
  6:     Predict $K_v \leftarrow \mathcal{F}(I)$
  7:     Compute loss $L \leftarrow \text{BCE}(y, K_v)$
  8:     Update $\mathcal{F}$ using gradient descent
  9: **end while**

---

**Algorithm 2** Watermark Embedding

---

**Require:** Original image $I_o$, encoder $\mathcal{E}$, mean $\mu$, covariance $\Sigma$, iterations $T$, loss weights $\lambda_D, \lambda_W$
**Ensure:** Watermarked image $I_w$
  1: Initialize $I_w \leftarrow I_o$
  2: **for** $t = 1$ to $T$ **do**
  3:     Compute distortion loss: $L_D \leftarrow -\frac{1}{2}(\mathcal{E}(I_w) - \mu)^\top \Sigma^{-1}(\mathcal{E}(I_w) - \mu)$
  4:     Generate binary label: $y \leftarrow \{\text{sign}(\tilde{\mathcal{E}}(I_w) - \mu)\}_{\{i_1, \ldots, i_m\}}$
  5:     Compute softened label: $\tilde{y} \leftarrow \sigma(y)$          ▷ Sigmoid activation
  6:     Calculate watermark loss: $L_W \leftarrow S_\theta(I_w, \tilde{y})$
  7:     Total loss: $L \leftarrow \lambda_D L_D + \lambda_W L_W$
  8:     Compute perturbation $\eta_t$ using MI-FGSM: $\eta_t \leftarrow \text{MI-FGSM}(\nabla_{I_w} L)$
  9:     Update image: $I_w \leftarrow \text{Clip}(I_w + \eta_t)$
10: **end for**

---

## E  WATERMARKING ALGORITHM DETAILS

Our proposed watermarking framework consists of three key components: (1) score estimator pre-training, (2) watermark embedding through adversarial perturbation, and (3) watermark detection. The complete pseudocode is provided in Algorithms 1-3.

## F  WATERMARK VISUALIZATION

We provide more visualizations of the watermark to illustrate its characteristics. As shown in Figure 8, the watermark embedded in the images exhibits a strong correlation with the image content itself, ensuring seamless integration while maintaining imperceptibility.

We also include visual comparisons of the 10× amplified watermark deltas between original and watermarked images for both baseline methods and our proposed approach in Figure 9.

## G  ADDITIONAL DISTORTION ANALYSIS RESULTS

We provide additional results comparing the distortion introduced by different watermarking methods in the output of the latent diffusion model with the prompt *Generate an image in the impressionism style of the original image*. Figure 10 to Figure 14 illustrate the generated images for various watermarking techniques.

We also performed an evaluation on generating Expressionism-style outputs and obtained an FID score of 162.16 using our method, compared to 95.09–105.60 for the baselines, indicating the effectiveness of our method across different artistic styles. A corresponding visualization is shown in Figure 15.

---

**Algorithm 3** Watermark Detection

---

**Require:** Suspected image $I_s$, significance level $\alpha$, calibration distribution scores $\{s_i\}_{i=1}^N$
**Ensure:** Detection decision
 1: Retrieve $y$ from training phase
 2: Compute test statistic: $s_s \leftarrow S_\theta(I_s, y)$
 3: Calculate empirical p-value: $\hat{q}(s_s) \leftarrow \frac{1}{N} \sum_{i=1}^N \mathbf{1}(s_i \leq s_s)$
 4: **if** $\hat{q}(s_s) < \alpha$ **then**
 5:     **return** "Reject $H_0$ (Watermarked)"
 6: **else**
 7:     **return** "Retain $H_0$ (Not watermarked)"
 8: **end if**

---

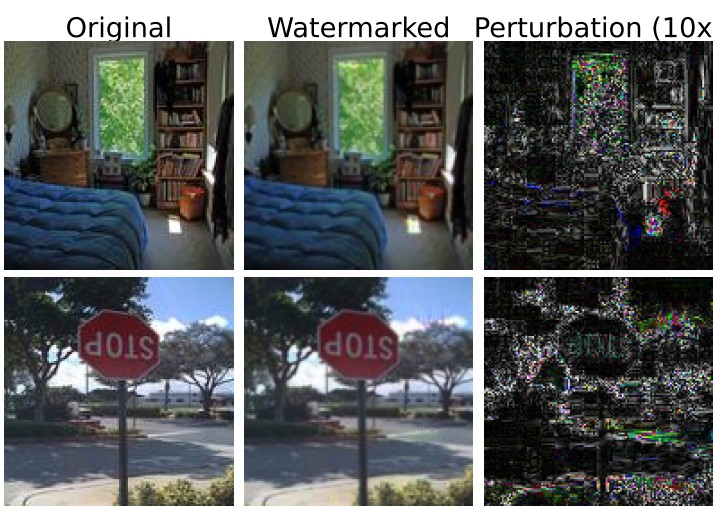

Figure 8: Visual comparison of original images, watermarked images, and 10× amplified structural differences.

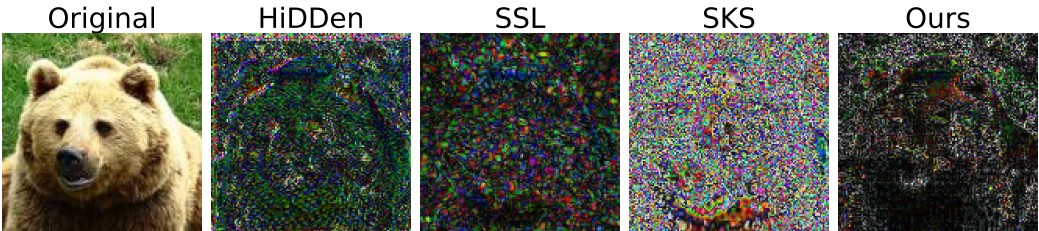

Figure 9: Visual comparison of watermark patterns across different methods. (Left) Original image; (Right) Corresponding 10× amplified watermark deltas.

## H   USE OF LARGE LANGUAGE MODELS

We used large language models to assist with editing and polishing manuscript text (e.g., wording, grammar, and clarity). Suggestions from large language models were reviewed, modified, and approved by the authors. The authors retain full responsibility for the final content, data interpretation, and conclusions.

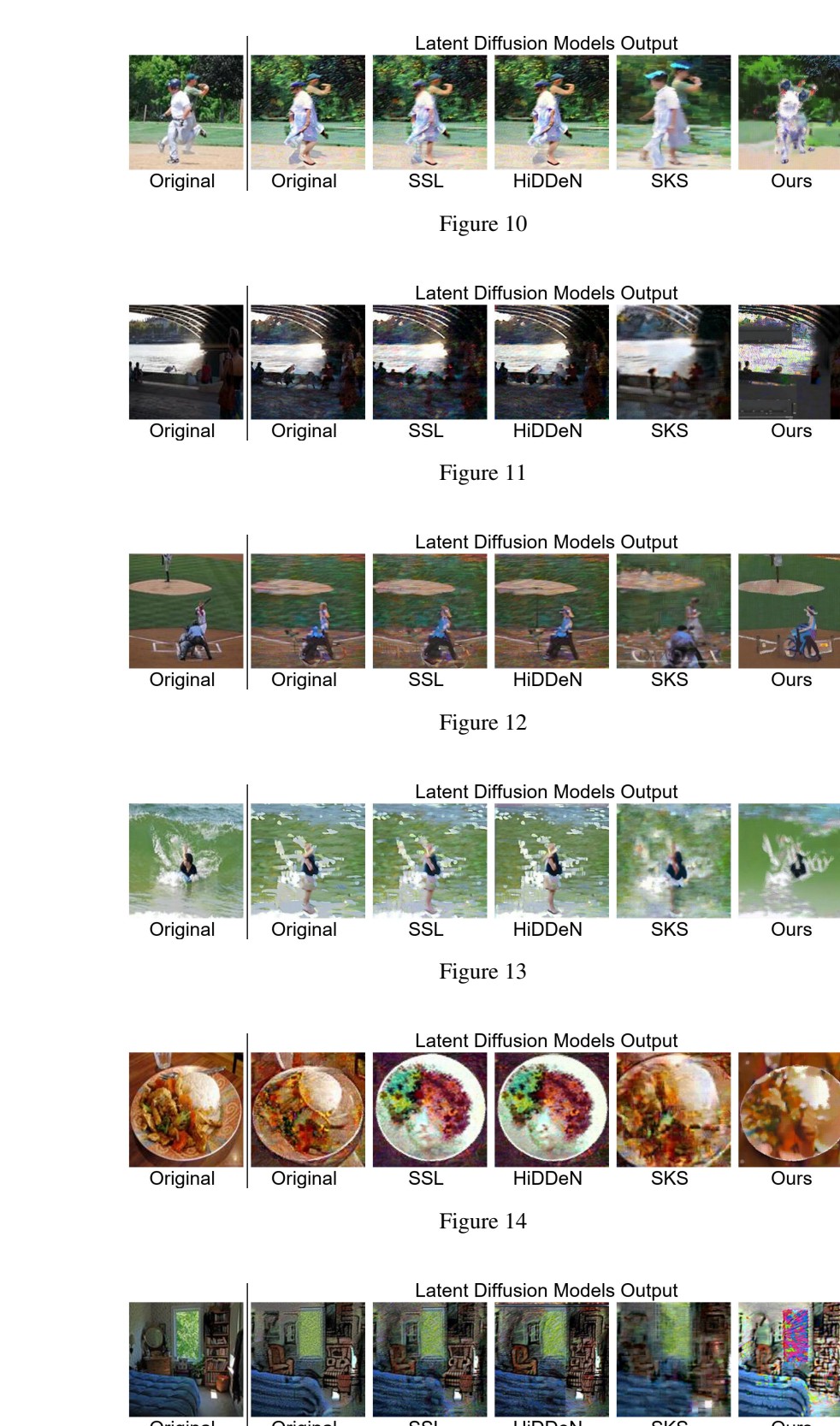

Figure 10

Figure 11

Figure 12

Figure 13

Figure 14

Figure 15

