# OpenReview forum: "A Dual-Protection Framework for Copyright Protection and Image Editing Using Multi-Label Conformal Prediction"
_ICLR.cc/2026/Conference — ICLR 2026 Conference Withdrawn Submission_

### Official Review · Reviewer_NLZm · 2025-10-25

**Soundness:** 2
**Presentation:** 2
**Contribution:** 2
**Rating:** 4
**Confidence:** 4

**Summary:**

The paper introduces an image watermarking method that simultaneously addresses copyright protection and re-generation prevention. An image protected by this method carries a verifiable watermark, whose presence could be detected statistically based on multi-label conformal prediction. When the image is used for re-generation, the results would be visibly distorted. The authors evaluated this framework on multiple aspects and also justified certain design choices.

**Strengths:**

-	The paper proposes a unified framework to solve two critical problems: adding a watermark for ownership verification and preventing unauthorized re-generation of that image. This dual-functionality has practical significance.
-	The experimental evaluations are comprehensive, demonstrating the method’s performance across both tasks, its imperceptibility, generalization capabilities, robustness, and security, followed by ablation studies on key design choices.

**Weaknesses:**

-	Regarding watermark detection, the claim of achieving “comparable” detection rates to baseline methods (line 292) is not directly supported by experiment results.
-	Regarding protection against re-generation, the method shows superior performance over adapted watermarking techniques but is not compared against methods specifically designed for this purpose (e.g., Glaze, AdvDM). Involving comparisons against these methods could make the evaluation more fair and convincing.
-	As acknowledged by the authors, when protecting against unauthorized generation, the method has limited cross-model generalizability.
-	As a 0-bit design, the watermark’s robustness lacks behind state-of-the-art, especially under geometric transformations. Cropping with parameter 0.9 or a 10-degree rotation could cause the accuracy rate drop below 40%.

By addressing these issues (mainly the first two), the paper’s main claims could be better supported.

**Questions:**

-	Compared to baseline methods selected in the paper, would the proposed per-image optimization process require more time to watermark an image?
-	The Score Estimator is a separate network trained on labels derived from the LDM's VAE encoder. Could this be considered a form of knowledge distillation? If so, what was the motivation for training a new model rather than more directly using the VAE encoder itself to derive the conformity score?

---

### Official Review · Reviewer_qxif · 2025-10-30

**Soundness:** 2
**Presentation:** 2
**Contribution:** 2
**Rating:** 2
**Confidence:** 4

**Summary:**

The paper proposes a dual protection framework which embed imperceptible adversarial perturbations into images that functions both as a digital watermark and as a defense mechanism against LDMs. The method trains a score estimator on LDM latent embeddings and applies conformal inference to compute statistically valid p-values for watermark detection, while shifting image embeddings toward low-density latent regions to distort unauthorized generations. Experiments demonstrate the effectiveness of this approach in achieving good watermark detectability, visual imperceptibility, and resilience to diffusion-based editing.

**Strengths:**

1. The paper focuses on an important and interesting topic: traditional watermarks can be easily removed or degraded through diffusion-based image-to-image editing. It solves the problem by introducing a dual protection framework that embeds imperceptible perturbations into images, serving both as a verifiable watermark and as a mechanism to disrupt diffusion-based editing.
2. The proposed method is intuitively reasonable and demonstrated to be effective in disrupting editing.

**Weaknesses:**

1. The motivation for the dual-protection framework needs stronger justification as there have been multiple works which focuses on proposing watermarking method robust to image editing techniques such as [1][2].

[1] Robust Watermarking Using Generative Priors Against Image Editing: From Benchmarking to Advances

[2] Robust-Wide: Robust Watermarking against Instruction-driven Image Editing

2. The evaluation is not comprehensive enough. The main experiments are conducted with only one dataset and only one prompt for image-to-image editing, which is insufficient to demonstrate the effectiveness of the method across different settings. In addition, it is suggested that the authors should also demonstrate the performance of performance of baselines in Table 1 to show that whether the proposed method is able to achieve watermark detector accuracy similar to other methods.
3. The method fails to demonstrate good transferability when the score estimator is trained with one model and the watermarked images are input into another diffusion model. This raise a concern of the applicability of the method. In real practice, the owner of the images will not have the knowledge about what kind of diffusion model will be used for the attack.
4. While the proposed method demonstrates strong robustness against common attacks such as cropping and Gaussian noise, there exist more specialized and targeted watermark removal techniques, such as [3][4][5]. It is therefore suggested that the authors evaluate their method against these advanced removal attacks to provide a more comprehensive assessment of its robustness and practical reliability.

[2] Invisible Image Watermarks Are Provably Removable Using Generative AI

[3] Generative autoencoders as watermark attackers: Analyses of vulnerabilities and threats

[4] Evading watermark based detection of AI-generated content

5. In Section 5.5, the author demonstrate that the entire watermark pipeline can be transferred to another diffusion model, but only one model is considered in the experiments and no visualization are shown, which is not sufficient.

**Questions:**

1. Section 5.6 demonstrate that the proposed method is robust to several image corruption by showing the watermark detection rate after the corruption. Can the effect of the resilience to image editing also be retained after the attacking?
2. Is the proposed method able to achieve a comparable performance in defending malicious image editing with some adversarial method against image editing such as [1][2]?

[1] DiffusionGuard: A Robust Defense Against Malicious Diffusion-based Image Editing

[2] DCT-Shield: A Robust Frequency Domain Defense against Malicious Image Editing

3. If the watermark detector for the proposed method is applied to images watermarked by other watermarking methods such as HiDDeN, what will be the false negative rate?

4. Is there a trade-off between the watermark detection accuracy and the effect of defending malicious image editing?

---

### Official Review · Reviewer_PsPo · 2025-10-31

**Soundness:** 2
**Presentation:** 2
**Contribution:** 2
**Rating:** 2
**Confidence:** 4

**Summary:**

This paper proposes a dual approach for watermark protection. It embeds an imperceptible perturbation in images, serving as a watermark while simultaneously disrupting the output of latent diffusion models. Experiments demonstrate the effectiveness of the proposed method.

**Strengths:**

[1] The paper is clear and easy to understand, clearly stating the motivation and the intuition.

[2] Experiments are performed to demonstrate the effectiveness of the method.

[3] Copyright protection is an important topic.

**Weaknesses:**

[1] My major concern is the lack of justification of the novelty of the proposed work. The major issue is that the authors only includes two papers in 2024 and there is no paper in 2025. The missing of the references could prevent authors from understanding the novelty and the importance of the paper. While there may be not many papers when considering the specific techniques, many papers are published in this field and should be acknowledged. The following list a few literature:

Bui, Tu, Shruti Agarwal, and John Collomosse. "TrustMark: Robust Watermarking and Watermark Removal for Arbitrary Resolution Images." Proceedings of the IEEE/CVF International Conference on Computer Vision. 2025.

Liu, Gaozhi, et al. "Watermarking One for All: A Robust Watermarking Scheme Against Partial Image Theft." Proceedings of the Computer Vision and Pattern Recognition Conference. 2025.

Wu, Shaowu, Wei Lu, and Xiangyang Luo. "Robust Watermarking Based on Multi-layer Watermark Feature Fusion." IEEE Transactions on Multimedia (2025).

Dzhanashia, Kristina, and Oleg Evsutin. "Robust image watermarking for diverse channels with template-forming neural network." Applied Soft Computing (2025): 114125.

Xu, Rui, et al. "InvisMark: Invisible and Robust Watermarking for AI-generated Image Provenance." 2025 IEEE/CVF Winter Conference on Applications of Computer Vision (WACV). IEEE, 2025.

The authors are suggested to revise the introduction and related works section about this. And experiments should also compare with more recent methods as well.

[2] The robustness evaluation only considers "Gaussian noise, Gaussian blur, cropping, rotation, and JPEG compression", but not those more adversarial edits. The authors are suggested to add experiments for other techniques, e.g. Bui et al. (2025).

[3] In "Case 1: Can watermarked images trained on one diffusion model withstand attacks from another?", the limitation on the transferability could be a concern of the proposed approach. I'm wondering if the authors could provide a mitigation strategy for it (intuition is enough). The transferability is important when applying watermark techniques in real practice.

**Questions:**

Please address my concerns in the above.

---

### Official Review · Reviewer_JgXK · 2025-11-01

**Soundness:** 3
**Presentation:** 2
**Contribution:** 2
**Rating:** 4
**Confidence:** 3

**Summary:**

The paper proposes a dual-protection scheme that adds an imperceptible, adversarial perturbation to images which (i) acts as a watermark detectable via multi-label conformal prediction on a learned "score estimator" and (ii) shifts latent embeddings away from the data distribution so that latent diffusion models produce distorted edits. In the evaluation part, this paper contrasts this design with prior key-based watermarks and reports results covering detection, imperceptibility, distortion on edits, robustness to common transforms, and several security scenarios.

**Strengths:**

- The paper addresses a timely and important problem of protecting image copyrights amid diffusion-based editing and regeneration.
- The dual-protection scheme is novel.
- The reported results suggest the method achieves watermark detectability and editing deterrence while maintaining imperceptibility across evaluated scenarios.

**Weaknesses:**

- The "distortion" evaluation relies primarily on FID computed between edited outputs and reference art-style images, which is known to conflate fidelity/diversity and to be sensitive to the feature backbone. Alternative metrics (e.g., precision/recall decompositions) and/or human studies would strengthen the claims [1].
- The security section focuses on three fixed scenarios (fake watermarking, direct claims, and removal/replacement), but there's no experiment where an attacker adapts to this specific detector or uses detector feedback to tune an attack. This matters because defenses that look fine under fixed tests often fail once attacks are tailored to them, and even simple, query-efficient black-box methods can succeed with limited feedback [2,3,4]. The method also relies on the owner’s secret choice of latent dimensions staying hidden (Section 3.1.1), but there is no analysis of how much a probing attacker could infer from scores or p-values.
- The claim that a moderate-accuracy VGG backbone "provides better uncertainty calibration" for conformal prediction (Section 5.1) is asserted without calibration evidence.
- The paper itself shows that watermarks trained with one diffusion model do not reliably distort edits from different models (Section 5.5, Case 1), and only report success after retraining the pipeline for each target model (Case 2), which increases deployment cost and narrows protection unless such retraining is feasible at scale.

### References
- [1] G. Stein et al. Exposing Flaws of Generative Model Evaluation Metrics and Their Unfair Treatment of Diffusion Models. NeurIPS 2023.
- [2] A. Athalye et al. Obfuscated Gradients Give a False Sense of Security. ICML 2018.
- [3] F. Tramèr et al. On Adaptive Attacks to Adversarial Example Defenses. NeurIPS 2020.
- [4] A. Ilyas et al. Black-box Adversarial Attacks with Limited Queries and Information. ICML 2018.

**Questions:**

- The security model hinges on secret selection of m latent dimensions (Section 3.1.1); can an adaptive adversary with query access infer these dimensions, and how does this compare formally with SKS-style secret keys?
- Section 5.1.1 states VGG yields “better uncertainty calibration” for conformal embedding. Can you elaborate more on this with theoretical or empirical results?
- Section 5.5 shows limited cross-model transfer; what practical deployment path do you envision: per-encoder retraining or a mechanism to improve transfer?

---

### Note · Authors · 2025-11-16

I have read and agree with the venue's withdrawal policy on behalf of myself and my co-authors.